# LLMGD: Compressing LLMs with Layerwise Geodesic Distances

## Abstract

Understanding how internal representations evolve across layers in large language models (LLMs) is critical for interpretability, robustness, and efficient model design. We introduce LLMGD, a stability-guided pruning metric grounded in spectral graph theory. For each layer, we estimate a precision matrix from embedding vectors that characterize the model's internal states, and then compute the geodesic distance on the cone of symmetric positive definite (SPD) matrices between successive layers. This yields a smooth and robust measure of representational distortion, identifying layers with minimal geodesic change as candidates for removal or replacement, thereby providing a principled foundation for model compression. Empirically, across multiple LLMs and tasks, including OPT-1.3B and OPT-2.7B models, LLMGD consistently detects structurally redundant segments and, when combined with lightweight replacement layers, delivers strong compression–accuracy trade-offs compared to existing pruning methods. We further establish a bi-Lipschitz upper-bound interpretation of LLMGD, which clarifies its robustness as a pruning criterion. Together, these results demonstrate that LLMGD reliably identifies structurally important layers and enables robust model compression with minimal performance degradation.

## 1 Introduction

Large Language Models (LLMs) have emerged as foundational tools in natural language processing, achieving impressive performance across a wide range of tasks, including text generation, translation, code synthesis, and reasoning (Vaswani et al., 2017; Touvron et al., 2023; Zhang et al., 2022). However, their increasing scale and complexity come at the cost of substantial computational demands and reduced interpretability. Gaining a clear understanding of the functional role of each layer is essential to effectively address these challenges. While earlier work often assumed that all layers contribute equally, recent studies suggest layers differ significantly in their contribution to model performance (Geva et al., 2021; Georges Gabriel Charpentier & Samuel, 2023; Voita et al., 2019a; Chen et al., 2025; Yang et al., 2024). This observation motivates the development of principled and interpretable methods to assess the functional significance of transformer layers.

Layers of LLMs can be viewed as complex functions that map input tokens to high-dimensional vectors, commonly referred to as hidden state embeddings. The input and output embeddings of each layer, computed over a batch of data, can serve as indicators of the functional behavior of that layer. By analyzing how the geometry of the input space is transformed into the output space, and vice-versa, one can infer both the importance and potential redundancy of a layer. This insight is valuable for several downstream applications, including model pruning (Chen et al., 2025), dynamic layer selection (Dotzel et al., 2024), and robustness to noise. This motivates geometric criteria that directly capture representational change across layers.

A central challenge in understanding large neural architectures lies in the absence of mathematically rigorous tools to quantify how information is distorted across layers. While empirical heuristics for layer importance exist, they often lack a theoretical foundation. To address this, we introduce **LLMGD**, a layerwise criterion based on geodesic distances between precision matrices estimated from hidden states. Specifically, we use probabilistic graphical models (PGMs) to derive precision matrices that summarize variable dependencies within each layer's embeddings. These matrices are symmetric positive definite (SPD), and thus naturally lie on a non-Euclidean geometry where

distances can be measured using the affine-invariant Riemannian metric (AIRM). By computing geodesic distances between successive layers, we obtain a smooth, scale-aware measure of representational distortion. Layers inducing minimal change can be identified as candidates for removal or lightweight replacement, enabling stability-guided pruning.

In later sections, we establish a theoretical connection between this geodesic distance metric and the bi-Lipschitz constant, a classical measure of how much a function can stretch or compress distances. We show that LLMGD provides an upper bound on bi-Lipschitz distortion, thereby grounding our pruning signal in a well-established theoretical framework.

Finally, we compare our metric with existing approaches for quantifying layer importance in model pruning. Empirical results demonstrate that it effectively distinguishes between essential and redundant layers, achieving substantial improvements at equivalent pruning ratios compared to state-of-the-art metrics (Chen et al., 2025), particularly in LLMs with 1.3B and 2.7B parameters.

Overall our key contributions are as follows:

- We present LLMGD, a novel layer wise criterion based on geodesic distances between SPD precision matrices summarizing hidden states.
- We demonstrate that LLMGD effectively captures functional distortion across layers, distinguishing essential from redundant components of transformer depth.
- Through pruning experiments, we show that LLMGD outperforms existing layer pruning metrics, delivering strong compression–accuracy trade-offs.
- We establish a bi-Lipschitz upper-bound interpretation of LLMGD, providing theoretical support for its robustness.

## 2 RELATED WORKS

### PROBABILISTIC GRAPHICAL MODELS

Probabilistic Graphical Models (PGMs) offer a principled framework for modeling complex dependencies among a set of random variables through graph-based representations (Koller & Friedman, 2009). In recent literature, PGMs have been applied to structured data scenarios where each node presents a sample, such as an embedding from a machine learning model and edges reflect statistically significant interactions or similarities. By representing joint distributions of high-dimensional data using graph-based factorizations, PGMs enable both interpretability and efficient inference. Studies have shown that PGMs successfully capture not only local, but global dependencies between these variables as well (Feng, 2021; Vu & Thai, 2020). Edges are formed between nodes that exhibit strong proximity or structural similarity, collectively capturing an approximation of the data's underlying manifold. Each node maintains a local neighborhood that encapsulates conditional dependencies, while the overall edge configuration reflects the global topological structure of the data manifold (Rubin-Delanchy, 2020). Dense subgraphs indicate clusters of high intrinsic coherence, often corresponding to regions with low variability, whereas sparse or weakly connected areas suggest increased uncertainty or diversity in the underlying distribution.

## 3 METHODOLOGY

### 3.1 OVERVIEW OF LLMGD

Figure 1 outlines the LLMGD pipeline, consisting of two phases: (Phase 1) **Precision Matrix Estimation** from hidden-state embeddings using probabilistic graphical models, and (Phase 2) **Geodesic Distance Calculation** between successive layers on the SPD manifold.

### 3.2 PHASE 1: PRECISION MATRIX ESTIMATION VIA PROBABILISTIC GRAPHICAL MODELS

**Hidden State Embedding Extraction & Pooling** To analyze the internal representations of a language model, we extract hidden-state vector embeddings from each transformer layer. Specifically, we randomly sample a subset of $N$ input sequences (tokens) from the dataset and pass them through

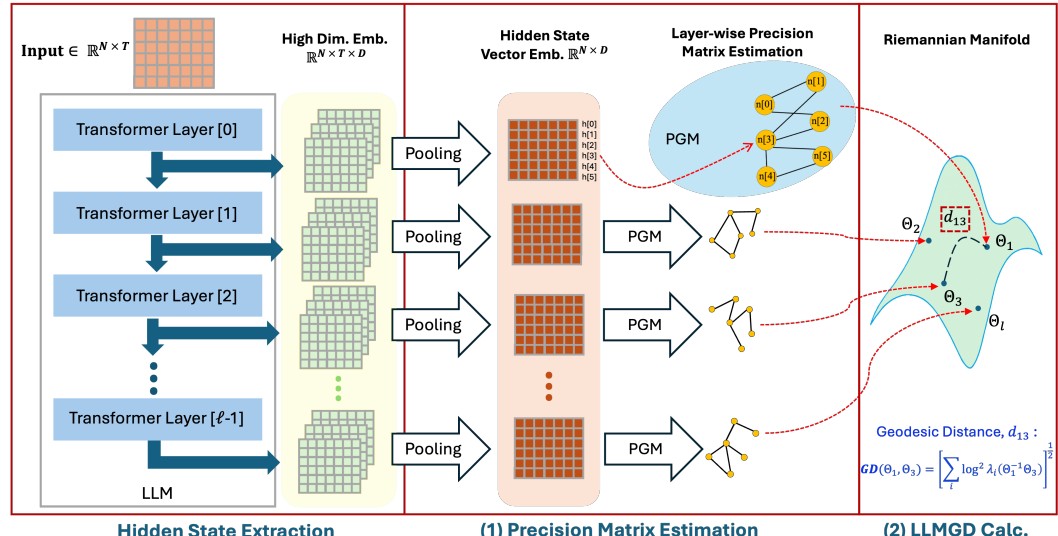

Figure 1: A high-level overview of the LLMGD pipeline, including hidden state extraction, (Phase 1) precision matrix estimation using PGM, and (Phase 2) geodesic distance computation between precision matrices using Riemannian metric.

the model in evaluation mode. For each input sample, we retrieve the full stack of hidden-states from all layers of the model. To obtain a fixed-size vector per layer, we apply mean pooling over the token dimension.

$$\mathbf{h}^{(\ell)} = \frac{1}{T} \sum_{t=1}^{T} \mathbf{H}_t^{(\ell)} \tag{1}$$

here $\mathbf{H}_t^{(\ell)} \in \mathbb{R}^D$ is the embedding of the $t$-th token at layer $\ell$ and $\mathbf{h}^{(\ell)} \in \mathbb{R}^D$ is the resulting mean-pooled vector representing the sequence at that layer.

By applying this to all $N$ samples in the dataset, we construct a matrix $\mathbf{P}^{(\ell)} \in \mathbb{R}^{N \times D}$, where each row corresponds to the pooled hidden state vector for that sample at layer $\ell$. Stacking these across all $L$ layers yields a tensor of the size $\mathbb{R}^{L \times N \times D}$, which is used in our analysis of layer geometry and structure.

**Precision Matrix Estimation** We consider a random vector $\mathbf{x} \sim \mathcal{N}(0, \Sigma)$ with the following probability density function:

$$f(\mathbf{x}) = \frac{\exp\left(-\frac{1}{2}\mathbf{x}^\top \Sigma^{-1} \mathbf{x}\right)}{(2\pi)^{N/2} \det(\Sigma)^{1/2}} \propto \det(\Theta)^{1/2} \exp\left(-\frac{1}{2}\mathbf{x}^\top \Theta \mathbf{x}\right) \tag{2}$$

where $\Sigma$ denotes the covariance matrix and $\Theta = \Sigma^{-1}$ is the corresponding precision matrix (the inverse covariance matrix).

Let $\mathbf{P} \in \mathbb{R}^{N \times D}$ denote the embedding matrix of a specific layer $(\ell)$, with each row representing a pooled hidden state for a sample. Our goal is to estimate a precision matrix $\Theta$ from this high-dimensional data, rather than from sample $\mathbf{x}$.

The graphical Lasso method (Friedman et al., 2008) aims at estimating the precision matrix by solving the following convex optimization problem that maximizes the log-likelihood of $f(\mathbf{x})$:

$$\max_{\Theta} : F(\Theta) = \log \det(\Theta) - \mathrm{Tr}\langle \Theta S \rangle - \beta \|\Theta\|_1 \tag{3}$$

where $\Theta$ denotes the precision matrix, $S$ denotes the sample covariance matrix, and $\beta$ is a regularization parameter. The sample covariance matrix $S$ is computed from $N$ independent and identically distributed (i.i.d.) samples $\mathbf{P} = [P_0, P_1, ..., P_{n-1}]$, where $\mathbf{P} \sim \mathcal{N}(0, S)$ is a zero-mean multivariate

Gaussian distribution in $\mathbb{R}^N$. Each element $\Theta_{i,j}$ in the precision matrix encodes the conditional dependencies between variables $P_i$ and $P_j$. For example, if $\Theta_{i,j} = 0$, then $P_i$ and $P_j$ are conditionally independent given all other variables.

To improve computational efficiency in high-dimensional settings, graph Laplacian estimation methods (Dong et al., 2019; Lake & Tenenbaum, 2010) have recently been introduced to solve the following optimization problem:

$$\max_{\Theta} : F(\Theta) = \log \det(\Theta) - \frac{1}{M} \text{Tr} \left( X^{\top} \Theta X \right) - \beta \|\Theta\|_1 \tag{4}$$

Subject to $\Theta = \mathcal{L} + \frac{1}{\sigma^2} I$, where $\mathcal{L}$ represents the set of valid Laplacian matrices. Here, $\text{Tr}(\cdot)$ denotes the matrix trace, $I$ is the identity matrix, and $\sigma^2 > 0$ represents the prior variance of the features. The three terms in Equation 4 correspond to $\log \det(\Theta)$, $\text{Tr}(\Theta S)$, and $\beta \|\Theta\|_1$ from the original graphical Lasso objective in Equation 3, respectively. When every row vector in the data matrix $\mathbf{P}$ is interpreted as a graph signal, there exists a close connection between the formulation in Equation 4 and the original graphical Lasso formulation (Friedman et al., 2008). In this formulation, all off-diagonal entries of $\Theta$ are non-positive, which leads to the estimation of attractive Gaussian Markov Random Fields (GMRFs) (Dong et al., 2019; Slawski & Hein, 2015). The relationship between the resulting precision matrix and the graph-based representation of the underlying data manifold is further explored in Section 4.

### 3.3 PHASE 2: GEODESIC DISTANCE CALCULATION LEVERAGING PRECISION MATRICES

We observe from Equation 4 that $\Theta$ is the sum of a graph Laplacian matrix $\mathcal{L}$ and a positive scalar multiple of the identity matrix. This operation effectively adds a positive constant of $\frac{1}{\sigma^2}$ to each diagonal element of $\mathcal{L}$, ensuring that the resulting precision matrix $\Theta$ is a SPD matrix. The LLMGD metric can be formally defined as the infimum length of geodesics connecting two data points in the Riemannian manifold formed by the cone of these SPD precision matrices (Lim et al., 2019). This can be imagined as a matrix representation of the geometric distance $|\log(a/b)|$ between two positive numbers $a, b$ (Bonnabel & Sepulchre, 2010; Shamai & Kimmel, 2017; Owen & Provan, 2010; Shuvo et al., 2024).

$$LLMGD(\Theta_1, \Theta_2) = \left[ \sum_{i=1}^{n} \log^2(\lambda_i(\Theta_1^{-1}\Theta_2)) \right]^{1/2} \tag{5}$$

where $\Theta_1$ and $\Theta_2$ are the precision matrices derived from the hidden state embeddings $\mathbf{P}_1$ and $\mathbf{P}_2$, respectively, and $\lambda_i$ is the $i$-th generalized eigenvalue obtained from the matrix pencil $(\Theta_1^{-1}\Theta_2)$.

The above formulation for computing distances between precision matrices is based on the Affine-Invariant Riemannian Metric (AIRM) (Lim et al., 2019). When LLMGD is used to compute the geodesic distance between two precision matrices, it is closely linked to the bi-Lipschitz maps between the two corresponding transformer layers, which will be discussed in the next section.

## 4 QUANTIFYING LAYERWISE BI-LIPSCHITZ MAPS IN LLMs WITH LLMGD

### 4.1 BI-LIPSCHITZ MAPS

In many representation learning tasks, it is desirable for a transformation to preserve the relative geometry of data points. Bi-Lipschitz maps formalize this idea by bounding how much distances can distort under the mapping.

A function $f : (X, d_X) \rightarrow (Y, d_Y)$ between two metric spaces is called $\kappa$-**bi-Lipschitz** if there exists a constant $\kappa \geq 1$ such that for all $p, q \in X$ and $f(p), f(q) \in Y$:

$$\frac{1}{\kappa} d_X(p, q) \leq d_Y(f(p), f(q)) \leq \kappa d_X(p, q) \tag{6}$$

Here $d_X(p, q)$ and $d_Y(f(p), f(q))$ denote the distances between points $p$ and $q$ points in the input space $X$ and output space $Y$, respectively.

Bi-Lipschitz maps are particularly useful in analyzing learned representations, as they preserve the relative distances between points in a stable and controlled manner. This makes them valuable for understanding embedding stability, robustness to perturbations, and geometry-preserving transformations in machine learning models.

**A Graph-Based Manifold Perspective on the Precision Matrix** The Laplacian component of the precision matrix encodes an undirected graph structure, which can be interpreted as a graph-based manifold. This manifold captures the conditional dependencies among vector embeddings: each node represents a data sample's embedding, while each edge encodes a pairwise conditional dependency. In practice, to estimate a graph structure corresponding to the precision matrix, we can first build a $k$-nearest neighbor ($k$-NN) graph over the vector embeddings and then use spectral graph sparsification to prune its edges, which is equivalent to the Maximum Likelihood Estimation (MLE) of the precision matrix. The detailed theoretical analysis has been provided in Appendix A.1 and A.2.

In the following, we show that the graph-based manifolds constructed in the previous step can be exploited to quantify the bi-Lipschitz constants of the maps between two transformer layers. Equation 6 implies that the bi-Lipschitz constant $\kappa$ satisfies:

$$\kappa := \max \left\{ \max_{\substack{p,q \\ p \neq q}} \frac{d_Y(p,q)}{d_X(p,q)}, \max_{\substack{p,q \\ p \neq q}} \frac{d_X(p,q)}{d_Y(p,q)} \right\} \tag{7}$$

where $p$ and $q$ are two data samples mapped from the input $X$ space to the output $Y$ space, and $d_X$, $d_Y$ denote their respective distance metrics on the graph-based manifolds.

**Theorem 4.1** (**LLMGD as an Upper Bound on Bi-Lipschitz Distortion**). *The LLMGD between two layers provides an upper bound on the logarithm of their bi-Lipschitz constant $\kappa$.*

*Proof.* To compute distances $d_X$ and $d_Y$ on graph-based manifolds, we consider effective-resistance distance to capture both local and global connectivity between nodes. Effective resistance, derived from spectral graph theory, draws an analogy between electrical networks and graphs, helping to quantify how easily current can flow between two nodes. Two nodes with lower effective-resistance distance between them imply higher connectivity (Ellens et al., 2011). Formally, for nodes $p$ and $q$, the effective resistance is computed as:

$$R_{\text{eff}}(p,q) = (\mathbf{e}_p - \mathbf{e}_q)^\top \mathcal{L}^\dagger (\mathbf{e}_p - \mathbf{e}_q), \tag{8}$$

where $\mathbf{e}_p$ and $\mathbf{e}_q$ denoting the standard basis vectors (i.e., vectors with a 1 at the $p$-th and $q$-th positions, respectively, and zeros elsewhere), and $\mathcal{L}^\dagger$ is the Moore-Penrose pseudoinverse (Barata & Hussein, 2012) of the graph Laplacian $\mathcal{L}$.

Since X and Y spaces can be represented as graph-based manifolds corresponding to their respective precision matrices, we can leverage resistance distance to quantify the bi-Lipschitz constant:

$$\max_{\substack{p,q \\ p \neq q}} \frac{d_Y(p,q)}{d_X(p,q)} = \max_{\substack{p,q \\ p \neq q}} \frac{R_{eff}^Y(p,q)}{R_{eff}^X(p,q)} = \max_{\substack{p,q \\ p \neq q}} \frac{\mathbf{e}_{pq}^\top \mathcal{L}_Y^\dagger \mathbf{e}_{pq}}{\mathbf{e}_{pq}^\top \mathcal{L}_X^\dagger \mathbf{e}_{pq}} \tag{9}$$

where $\mathbf{e}_{pq} = \mathbf{e}_p - \mathbf{e}_q$. As $\mathbf{e}_{pq}$ spans a subset of all vectors orthogonal to $\mathbb{1}$, we can upper-bound the above term by relaxing to all such vectors, leading to an expression involving generalized eigenvalues (Golub & Van Loan, 1996):

$$\max_{\substack{p,q \\ p \neq q}} \frac{\mathbf{e}_{pq}^\top \mathcal{L}_Y^\dagger \mathbf{e}_{pq}}{\mathbf{e}_{pq}^\top \mathcal{L}_X^\dagger \mathbf{e}_{pq}} \leq \max_{\substack{u \perp \mathbb{1} \\ u \neq 0}} \frac{u^\top \mathcal{L}_Y^\dagger u}{u^\top \mathcal{L}_X^\dagger u} = \max_{\substack{u \perp \mathbb{1} \\ u \neq 0}} \frac{u^\top \mathcal{L}_X u}{u^\top \mathcal{L}_Y u} = \lambda_{max}(\mathcal{L}_Y^\dagger \mathcal{L}_X) \approx \lambda_{max}(\Theta_Y^{-1} \Theta_X) \tag{10}$$

Since $\Theta_X$ and $\Theta_Y$ are SPD matrices, they are invertible, and their pseudoinverses are equal to their inverses (Stoer et al., 1980). Similarly, we have:

$$\max_{p \neq q} \frac{d_X(p,q)}{d_Y(p,q)} = \max_{p \neq q} \frac{R_{\text{eff}}^X(p,q)}{R_{\text{eff}}^Y(p,q)} = \max_{p \neq q} \frac{\mathbf{e}_{pq}^\top \mathcal{L}_X^\dagger \mathbf{e}_{pq}}{\mathbf{e}_{pq}^\top \mathcal{L}_Y^\dagger \mathbf{e}_{pq}}$$

$$\leq \max_{\substack{u \perp \mathbb{1} \\ u \neq 0}} \frac{u^\top \mathcal{L}_X^\dagger u}{u^\top \mathcal{L}_Y^\dagger u} = \max_{\substack{u \perp \mathbb{1} \\ u \neq 0}} \frac{u^\top \mathcal{L}_X u}{u^\top \mathcal{L}_Y u} = \lambda_{\max}(\mathcal{L}_X^\dagger \mathcal{L}_Y) \approx \lambda_{\max}(\Theta_X^{-1} \Theta_Y) \tag{11}$$

Since $\Theta_X$ and $\Theta_Y$ are both invertible matrices, we have $\lambda_{\max}(\Theta_X^{-1}\Theta_Y) = \frac{1}{\lambda_{\min}(\Theta_Y^{-1}\Theta_X)}$.

From Equation 5, we observe that the LLMGD metric incorporates all generalized eigenvalues between the two precision matrices in its computation. However, the extreme eigenvalues, the largest and smallest eigenvalues tend to contribute to the final value significantly. The eigenvalue spectra are shown in Figure 2, where the extreme eigenvalues represent a small but impactful portion of the overall distribution. As such, the LLMGD metric serves as an upper bound on the square root of the sum of squared logarithms of the extreme eigenvalues, thereby tightly bounding the bi-Lipschitz constant. This relationship is formalized in the following inequality:

$$LLMGD \geq \sqrt{\log^2(\lambda_{\max}) + \log^2(\lambda_{\min})} = \sqrt{\log^2(\lambda_{\max}) + \log^2(\frac{1}{\lambda_{\min}})} \geq \sqrt{\log^2(\kappa)} = \log(\kappa)$$
(12)

Here, $\lambda_{min}, \lambda_{max}$ are the generalized eigenvalues of the matrix pencil $(\Theta_1^{-1}\Theta_2)$. □

We observe that a smaller value of the LLMGD metric indicates a lower distortion between the input and output spaces (and vice versa), suggesting that this transformation preserves the geometric structure of the data.

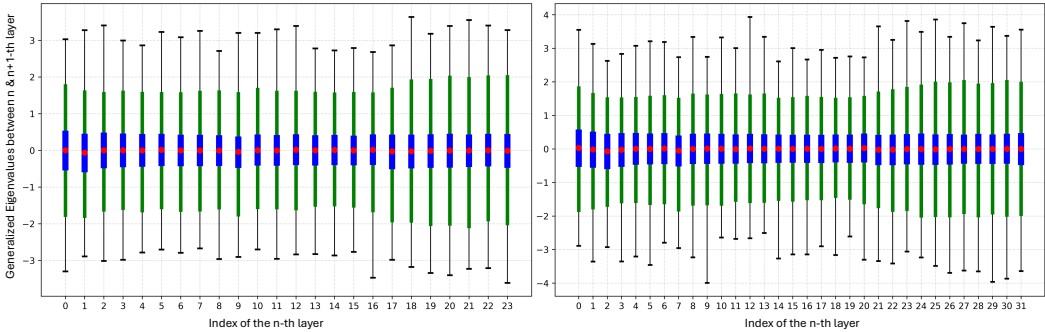

Figure 2: Generalized eigenvalues of input vs. output representations across OPT-1.3B and OPT-2.7B layers. Black: extreme 2 percentiles; green: outer quartiles (1st & 4th); blue: interquartile range (2nd & 3rd); red dot: median.The green envelopes for both OPT models are nearly identical, indicating that the token embeddings undergo similar geometric transformations across layers in comparable LLMs.

## 5 APPLICATIONS: STABILITY-GUIDED PRUNING & REPLACEMENT

Having established a principled metric for layerwise distortion, we now explore its utility in practical applications. One of the primary applications of our proposed method, LLMGD, is to identify structurally redundant layers within LLMs for layer-wise model pruning (Louizos et al., 2018; Chen et al., 2023; Frantar & Alistarh, 2023; Das et al., 2023; Sun et al., 2024; Xia et al., 2024; Ma et al., 2023; Chen et al., 2025). Traditional pruning techniques typically focus on removing specific components, such as attention heads (Michel et al., 2019; Voita et al., 2019b), filters (McCarley et al., 2019; Prasanna et al., 2020), or parameters that contribute to dimensionality (Xia et al., 2024; Hu et al., 2024; van der Ouderaa et al., 2024). In contrast, LLMGD offers a theoretically grounded alternative by quantifying the distortion between input and output manifolds at each layer.

Unlike recent studies that directly prune unimportant layers without any retraining (Song et al., 2024; Men et al., 2024), our method involves retraining a lightweight replacement model after pruning. This approach allows us to preserve the language model's functional capacity while reducing depth and computational complexity. Other approaches that incorporate fine-tuning after pruning (Yang et al., 2024; Kim et al., 2024; Gromov et al., 2025) demonstrate the importance of model adaptation post-pruning, an aspect that is useful for our application.

Since the geodesic distances derived from the AIRM reflect the degree of distortion introduced by each layer, they provide an effective criterion for identifying low-impact transformations. Layers exhibiting low LLMGD values correspond to those that preserve the global geometry of the data

space and thus contribute minimally to internal transformations. Such layers can be considered structurally redundant and pruned with minimal performance degradation.

Moreover, because LLMGD operates on internal hidden-state distributions, it generalizes across both architectures and datasets without requiring task-specific supervision. This enables a data-driven, bi-Lipschitz aware strategy for removing less informative layers in a principled way.

**Lightweight Replacement Layer**   In the lightweight training phase, a contiguous block of layers in the original model is replaced with a single trainable module. A group of $k$ consecutive layers is first identified as redundant based on similarity metrics and subsequently removed. A single layer is then initialized as a replacement and trained to approximate the input-output behavior of the removed layer block. To enable this, hidden-state pairs are extracted from the original model: specifically, the inputs to the first layer and the outputs of the last layer within the pruned segment. The replacement module is trained using these pairs to minimize the mean squared error (MSE) between its predictions and the true outputs. This process preserves the structural behavior of the original model while reducing both model depth and computational cost. Importantly, the replacement module retains both the multi-head self-attention and feedforward components, allowing it to serve as a compact yet expressive approximation of the pruned layer group.

For comparison, we also use two additional metrics: (1) Cosine similarity and (2) KL divergence- to evaluate the similarity of hidden-state distributions. These metrics are used to guide pruning while maintaining model performance and are briefly described in Appendix A.3 and A.4.

## 6   EXPERIMENTAL RESULTS

### 6.1   EXPERIMENTAL SETUP

Table 1: Classification accuracy of various pruning methods applied to the OPT-1.3B model across different pruning rates and classification benchmarks.

| Metric Used | Layers Pruned | Benchmarks on OPT-1.3B | | | |
|---|---|---|---|---|---|
| | | boolq | hellaswag | mmlu | piqa |
| Cosine Similarity | | 49.11 | **36.24** | 23.29 | 67.74 |
| KL_Divergence | 2 | 52.78 | 35.99 | 23.05 | 67.08 |
| LLMGD | | **54.95** | 35.70 | **25.05** | **68.39** |
| Cosine Similarity | | 57.71 | 29.71 | 23.05 | 62.51 |
| KL_Divergence | 4 | 56.24 | 30.59 | 23.06 | 63.06 |
| LLMGD | | **57.92** | **33.92** | **23.72** | **67.83** |
| Cosine Similarity | | 41.16 | 27.35 | **23.60** | 57.56 |
| KL_Divergence | 6 | 43.98 | 27.41 | 23.05 | 58.59 |
| LLMGD | | **59.97** | **30.43** | 22.97 | **62.35** |
| Cosine Similarity | | 37.83 | 26.95 | **23.47** | 58.05 |
| KL_Divergence | 8 | 52.81 | 25.39 | 23.24 | 53.26 |
| LLMGD | | **62.02** | **29.49** | 22.95 | **60.94** |
| Cosine Similarity | | 37.83 | 26.72 | **26.42** | 56.04 |
| KL_Divergence | 10 | 39.51 | 25.62 | 25.72 | 52.99 |
| LLMGD | | **61.49** | **28.60** | 22.92 | **60.01** |
| Baseline | N/A | 57.76 | 41.52 | 24.96 | 71.76 |

We conduct experiments on widely-used open-source large language models, including OPT-1.3B and OPT-2.7B (Zhang et al., 2022). Building on prior work in layer pruning and replacement (Dettmers et al., 2023; Chen et al., 2025), we evaluate multiple pruning configurations by varying both pruning rates and criteria. To identify optimal layers for pruning, we use 10,000 randomly selected samples from the RedPajama (Weber et al., 2024) corpus to compute three metrics: LLMGD (our proposed method), cosine similarity, and KL divergence across layers. These samples are used to construct the precision matrices for LLMGD computation, calculate pairwise cosine similarities between layer inputs and outputs, and measure KL divergence under perturbations. The resulting distributions across layers are illustrated in Appendix A.5. For training lightweight replacement modules that recover performance after pruning, we adopt domain-distribution sampling strategy (Xia et al., 2024). Specifically, we use 50,000 examples from RedPajama to train single Transformer layers that replace each pruned block, minimizing the mean squared error between the replacement module's output and

the original block's output. This larger training set ensures robust approximation of the removed layers' functionality while maintaining computational efficiency.

Table 2: Classification accuracy of various pruning methods applied to the OPT-2.7B model across different pruning rates and classification benchmarks.

| Metric Used | Layers Pruned | Benchmarks on OPT-2.7B | | | | | |
|---|---|---|---|---|---|---|---|
| | | arc_easy | boolq | copa | hellaswag | mmlu | piqa |
| Cosine Similarity | | 42.55 | 43.73 | **73.00** | 28.43 | **23.29** | 60.39 |
| KL_Divergence | 6 | 42.26 | **62.60** | 71.00 | **35.18** | 22.95 | 63.49 |
| LLMGD | | **48.23** | 60.46 | 68.00 | 31.57 | 22.99 | **65.94** |
| Cosine Similarity | | 33.25 | 41.68 | 58.00 | 26.58 | **24.90** | 56.04 |
| KL_Divergence | 10 | 34.18 | 38.78 | 61.00 | **28.58** | 22.95 | 57.45 |
| LLMGD | | **39.94** | **42.17** | **68.00** | 27.50 | 23.05 | **59.14** |
| Cosine Similarity | | 34.30 | 42.60 | 56.00 | 26.63 | **23.14** | 54.35 |
| KL_Divergence | 14 | 32.53 | 37.83 | 51.00 | 26.61 | 22.95 | 55.98 |
| LLMGD | | **39.94** | **51.59** | **66.00** | **28.35** | 22.91 | **60.39** |
| Cosine Similarity | | 29.76 | 37.89 | 55.00 | **26.47** | 23.10 | 54.73 |
| KL_Divergence | 18 | 29.55 | 37.83 | 52.00 | 26.23 | 22.95 | 54.13 |
| LLMGD | | **30.09** | **38.44** | **55.00** | 26.15 | **23.97** | **56.31** |
| Baseline | N/A | 60.77 | 60.24 | 77.00 | 45.85 | 25.61 | 73.78 |

## 6.2 PERFORMANCE EVALUATION

We evaluate the effectiveness of LLMGD on multiple downstream classification tasks under different pruning configurations. In each setting, a contiguous block of layers is removed and replaced with a lightweight module, guided by one of three metrics: LLMGD, Cosine Similarity, or KL Divergence. As shown in Tables 1 and 2, LLMGD consistently delivers competitive performance, often achieving the highest accuracy across tasks, and remaining close to the top when not achieving the highest score. Although its advantage varies across datasets and pruning ratios, LLMGD frequently outperforms the other metrics by a substantial margin, demonstrating its ability to effectively capture geometric distortion and identify structurally redundant layers with minimal impact on downstream performance.

As shown in Tables 1 and 2, pruning guided by LLMGD consistently produces higher or comparable classification accuracy compared to pruning based on cosine similarity or KL divergence. This trend holds across multiple pruning depths and benchmark datasets.

On the OPT-1.3B model, LLMGD outperforms both baselines in 16 out of 20 metric comparisons (across 4 benchmarks and 5 pruning depths). For example, at 10-layer pruning, LLMGD achieves 61.49% on BoolQ and 28.60% on HellaSwag, compared to 37.83% and 26.72% using cosine similarity. Even in cases where LLMGD does not attain the top score (e.g., MMLU at 10 layers pruning), it remains within a close margin, indicating strong robustness.

On OPT-2.7B, LLMGD continues to demonstrate superior performance under moderate pruning. For example, with six layers pruned, LLMGD achieves 48.23% on ARC-Easy and 65.94% on PIQA, outperforming cosine similarity by 5.68% and 5.55%, respectively. At 14 layers pruned, LLMGD improves COPA accuracy from 56.00% (cosine) to 66.00%, a 10-point gain, highlighting its capability to retain critical task-specific information even under aggressive compression.

Overall, LLMGD ranks highest in **16 out of 20 comparisons** on the OPT-1.3B model and **16 out of 24 comparisons** on OPT-2.7B, demonstrating its consistent advantage across architectures and pruning configurations. When not the top scorer, LLMGD remains highly competitive, closely matching cosine similarity and KL divergence.

This strong performance can be attributed to LLMGD' ability to measure nonlinear geometric distortion across hidden representations, providing a more accurate and globally-informed estimation

of representational redundancy than first-order metrics such as cosine similarity or KL divergence. These results validate LLMGD as a reliable and effective criterion for identifying structurally replaceable layers while preserving downstream task performance under aggressive compression.

### 6.3 RUNTIME AND COMPUTATIONAL COST

One of the strengths of the proposed LLMGD framework lies in its balance between theoretical rigor and computational feasibility. While computing geodesic distances between precision matrices on the SPD manifold involves matrix operations such as generalized eigenvalue decomposition, the overall computational cost remains feasible for practical sample sizes and model depths.

For a given layer $\ell$, we first compute the precision matrix $\Theta_\ell \in \mathbb{R}^{N \times N}$ from the hidden state embeddings using a PGM, where $N$ is the number of samples. This involves constructing a $k$-nearest neighbor graph over the embeddings, which has a near-linear time complexity of $\mathcal{O}(N \log N)$ (Cheng et al., 2024; Malkov & Yashunin, 2018), followed by spectral sparsification with complexity $\mathcal{O}(Ndm)$, where $d$ is the average degree and $m$ is the order of the Krylov subspace (Cheng et al., 2024). Once the precision matrices for two layers are obtained, the geodesic distance via the Affine-Invariant Riemannian Metric is computed using the generalized eigenvalues of the matrix pencil $(\Theta_1^{-1}\Theta_2)$. This eigenvalue decomposition

Table 3: Computation time for LLMGD between two layers at varying sample sizes.

| Number of Samples (N) | Avg Time of Computation |
|---|---|
| 1000 | 0.430s |
| 2000 | 2.907s |
| 5000 | 40.089s |
| 10000 | 313.21s |

step has a computational complexity of $\mathcal{O}(N^3)$ and constitutes the primary computational bottleneck in our pipeline. However, all previous steps are near-linear and thus scale efficiently with $N$. In practice, we further reduce computational overhead by applying mean-pooling over token embeddings before constructing the graph. The total runtime required to compute LLMGD for varying sample sizes is reported in Table 3.

## 7 LIMITATIONS

Although LLMGD provides a principled and effective approach to stability-guided model pruning, several aspects remain open for future exploration. Our current study focuses on medium scale models and classification benchmarks, leaving opportunities to extend the analysis to larger architectures and a wider range of downstream tasks such as reasoning, code generation, and long context understanding. Furthermore, evaluating LLMGD within broader model optimization pipelines, including dynamic pruning and quantization, offers a promising direction for integrating geometric signals with complementary compression strategies.

## 8 CONCLUSION

In this work, we proposed LLMGD, a geodesic-distance–based criterion for assessing layer importance in large language models. By modeling layer-wise hidden state distributions as precision matrices and computing distances on the SPD cone, our method provides a scalable and theoretically grounded measure of representational distortion. We demonstrated that LLMGD effectively identifies redundant layers and consistently outperforms similarity-based metrics such as cosine distance and KL divergence in pruning experiments. Combined with lightweight replacement modules, it yields strong compression–accuracy trade-offs, offering a principled framework for model optimization. Finally, we established a bi-Lipschitz upper-bound interpretation of LLMGD, which explains its robustness as a pruning signal and connects our empirical findings to a broader theoretical foundation.

## LLM USEAGE

We used LLM-based tools to improve the clarity of a few sentences and to correct grammatical errors.

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

# A    APPENDIX

## A.1    GRAPH-BASED MANIFOLD FORMATION

**k-nearest neighbor graph construction**    Given a set of pooled hidden-state embeddings $\{P_0, P_1, ..., P_{n-1}\} \in \mathbb{R}^{N \times D}$, we construct a $k$-NN graph $G = (\mathcal{V}, \mathcal{E})$, where each node $v_i \in \mathcal{V}$ corresponds to a data sample embedding ($P_i$), and edges are formed between a node and its $k$ most similar neighbors. Similarity is measured using the Euclidean distance in the embedding space.

The adjacency matrix $\mathbf{A} \in \mathbb{R}^{N \times N}$ is constructed such that $A_{ij} = 1$ if $\mathbf{v}_j \in \text{NN}_k(\mathbf{v}_i)$, and 0 otherwise. We symmetrize the graph by enforcing $A = \max(\mathbf{A}, \mathbf{A}^\top)$ to ensure an undirected structure.

## A.2    OBJECTIVE MAXIMIZATION VIA EDGE PRUNING

Following prior work (Cheng et al., 2024), it has been shown that maximizing the objective

$$\max_{\Theta} F(\Theta) = \log \det(\Theta) - \frac{1}{k} \text{Tr}(X^\top \Theta X), \tag{13}$$

where $\Theta = \mathcal{L} + \frac{1}{\sigma^2} I$, can be achieved by removing (or down-weighting) edges with small distance ratios. Such edges contribute little to the log-determinant term while incurring a disproportionately large penalty in the trace term, and their removal can increase the overall value of $F(\Theta)$.

**Objective Decomposition**    The Laplacian is expressed as

$$\mathcal{L} = \sum_{\substack{(p,q) \in E \\ p \neq q}} w_{pq} \mathbf{e}_{pq} \mathbf{e}_{pq}^\top,$$

The objective is decomposed in two terms as:

$$F(\Theta) = F_1(\Theta) - \frac{1}{k} F_2(\Theta),$$

where
$$F_1(\Theta) = \log \det(\Theta), \quad F_2(\Theta) = \text{Tr}(X^\top \Theta X).$$

Since $\Theta = \mathcal{L} + \frac{1}{\sigma^2} I$, each edge weight $w_{p,q}$ appears explicitly through the Laplacian $\mathcal{L}$.

**Gradient with Respect to an Edge Weight**    To analyze the effect of an individual edge, the gradient of $F(\Theta)$ with respect to $w_{p,q}$ is considered.

$F_1(\Theta)$:    Let $\lambda_i$ and $v_i$ denote the eigenvalues and eigenvectors of $\Theta$, respectively. By applying matrix calculus, it has been shown that:

$$\frac{\partial F_1}{\partial w_{pq}} = \frac{\partial}{\partial w_{pq}} \log \det \left( \mathcal{L} + \frac{1}{\sigma^2} I \right) \approx d_{\text{eff}}(p, q),$$

where $d_{\text{eff}}(p, q)$ represents the effective resistance between nodes $p$ and $q$, which captures how strongly edge $(p, q)$ influences $\log \det(\Theta)$.

$F_2(\Theta)$:

$$F_2(\Theta) = \text{Tr}(X^\top \Theta X) = \text{Tr}\left( X^\top \left( \mathcal{L} + \frac{1}{\sigma^2} I \right) X \right) = \frac{\text{Tr}\left( X^\top X \right)}{\sigma^2} + \sum_{(p,q) \in E} w_{pq} \left\| X^\top e_{pq} \right\|_2^2.$$

Using $\left\| \mathbf{X}^\top e_{pq} \right\|_2^2 = \mathbf{X}^\top \mathbf{e}_{pq} \mathbf{e}_{pq}^\top \mathbf{X} = \| \mathbf{X}_p - \mathbf{X}_q \|_2^2 = d_{\text{dat}}(p, q)$, the derivative becomes:

$$\frac{\partial F_2}{\partial w_{pq}} = \left\| \mathbf{X}^\top e_{pq} \right\|_2^2 = d_{\text{dat}}(p, q),$$

Since $d_{\text{dat}}(p, q) = \frac{1}{w_{pq}}$, it follows that:

$$\frac{\partial F_2}{\partial w_{pq}} = \frac{1}{w_{pq}}$$

So, the derivative of $F(\Theta)$ w.r.t. $w_{pq}$ is

$$\frac{\partial F}{\partial w_{pq}} = d_{\text{eff}}(p, q) - \frac{1}{k}\frac{1}{w_{pq}} = 0 \tag{14}$$

**Distance Ratio and Edge Ranking**  Rewriting Equation 14:

$$d_{\text{eff}}(p, q) = \frac{1}{k}\frac{1}{w_{pq}}$$

The pruning condition is reformulated using the *distance ratio*:

$$\rho_{p,q} = \frac{d_{\text{eff}}(p, q)}{d_{\text{dat}}(p, q)} = w_{pq}d_{\text{eff}}(p, q).$$

This ratio serves as a heuristic for edge importance:

- Large $\rho_{p,q}$: the edge is important to keep $\log\det(\Theta)$ high and should be retained.
- Small $\rho_{p,q}$: the edge contributes little and can be pruned.

**Pruning Strategy**  Edges with low $\rho_{p,q}$ values are removed to maximize $F(\Theta)$. This approach retains edges that preserve the spectral properties of the graph-based manifold via $\log\det(\Theta)$—while reducing the trace penalty induced by weakly informative but distant edges. As a result, a structurally faithful and more efficient representation is obtained.

SPECTRAL GRAPH SPARSIFICATION

Edges with low $\rho_{p,q}$ are candidates for pruning as they contribute less to maximizing the objective function $F(\Theta)$.

These retained edges preserve the spectral properties of the graph-based manifold via $\log\det(\Theta)$, while reducing the trace penalty from weakly informative connections. The ratio $\rho_{p,q}$ corresponds to the edge sampling probability used in spectral graph sparsification (Spielman & Teng, 2011). Spectral sparsification aims to approximate the original graph with a sparser one while preserving its spectral (Laplacian) properties. Here:

- Edges are sampled with probability proportional to $w_{pq}d_{\text{eff}}(p, q)$.
- Edges with higher $\rho_{p,q}$ are more likely to be included in the sparsified graph.

Therefore, our edge pruning strategy, performing spectral sparsification on the initial dense graph is equivalent to maximizing the objective function in Equation 4. To implement this efficiently, we adopt the Low-Resistance Diameter (LRD) decomposition scheme (Aghdaei & Feng, 2024; Cheng et al., 2024; Aghdaei & Feng, 2022), which approximates edge-wise effective resistance in weighted graphs, offering a viable and scalable path toward spectral sparsification. This ensures that the essential structural properties of the graph are preserved while significantly reducing complexity.

A.3  COSINE SIMILARITY

A widely adopted metric for identifying redundant or less important layers in LLMs is the cosine similarity (Chen et al., 2025) between a layer's input and output hidden states, which treats each Transformer decoder layer as a residual transformation applied to the input hidden states.

As a definition, for the $\ell$-th layer with parameters $\theta^{(\ell)}$ and input hidden state $\mathbf{p}^{(\ell)}$, the residual form of the transformation is given by:

$$\mathbf{p}^{(\ell+1)} = \mathbf{p}^{(\ell)} + f(\mathbf{p}^{(\ell)}, \theta^{(\ell)}), \tag{15}$$

where $f$ denotes the transformation function of the layer. If the output $\mathbf{p}^{(\ell+1)}$ is highly similar to the input $\mathbf{p}^{(\ell)}$, the transformation contributed by $f$ is minimal, and thus the layer may be considered redundant. To quantify this, cosine similarity is computed between $\mathbf{p}^{(\ell)}$ and $\mathbf{p}^{(\ell+1)}$ across randomly sampled sequences:

$$\cos\left(\mathbf{p}^{(\ell)}, \mathbf{p}^{(\ell+1)}\right) = \mathbb{E}_{(\mathbf{p}_i^{(\ell)}, \mathbf{p}_i^{(\ell+1)}) \in \mathcal{H}} \left[ \frac{1}{L} \sum_{j=1}^{L} \frac{\mathbf{p}_{i,j}^{(\ell)} \cdot \mathbf{p}_{i,j}^{(\ell+1)}}{\|\mathbf{p}_{i,j}^{(\ell)}\| \cdot \|\mathbf{p}_{i,j}^{(\ell+1)}\|} \right], \tag{16}$$

where $\mathcal{H}$ is the set of hidden state pairs from a set of samples. $\mathbf{p}_i^{(\ell)}, \mathbf{p}_i^{(\ell+1)} \in \mathbb{R}^{D \times L}$ denote the input and output hidden states of the $i$-th sample, respectively. $D$ denotes the hidden size, $L$ denotes the sequence length, and $j$ indexes over tokens.

For pruning, contiguous layers that yield the highest inter-layer cosine similarity over a fixed interval $n$ are selected, defined as:

$$\ell^*(n) = \arg\max_{\ell} \cos\left(\mathbf{p}^{(\ell)}, \mathbf{p}^{(\ell+n)}\right), \tag{17}$$

which identifies the least informative contiguous block of $n$ layers.

While cosine similarity provides a useful, lightweight metric for measuring redundancy, it captures only angular relationships in the embedding space. However, it fails to capture intrinsic distortions in the embedding manifold, which is a more global measure of layer behavior. We compare cosine similarity with our metric in Section 6.

### A.4 KL_DIVERGENCE

We assess the functional redundancy between layers by computing the Kullback-Leibler (KL) divergence (Hershey & Olsen, 2007; Pérez-Cruz, 2008) between their hidden state distributions. Let $\mathbf{H}^{(\ell)} \in \mathbb{R}^{T \times D}$ denote the output hidden state of layer $\ell$ for a given sample input, where $T$ is the number of tokens and $D$ is the hidden dimension.

To evaluate the impact of perturbations and assess layer stability, we add small Gaussian noise to the source layer $\ell$:

$$\widetilde{\mathbf{H}}^{(\ell)} = \mathbf{H}^{(\ell)} + \epsilon, \quad \epsilon \sim \mathcal{N}(0, \sigma^2 \mathbf{I}) \tag{18}$$

where $\sigma$ is a small positive constant (e.g., $\sigma = 10^{-2}$). This mimics random activation fluctuations and enables us to assess the sensitivity of downstream layers to perturbations in earlier representations.

We then propagate the perturbed hidden states through the remaining layers and compare the output at the target layer $\ell + k$. For each token position $t \in \{1, \ldots, T\}$, we normalize the hidden states into token-wise distributions using the softmax function:

$$\mathbf{P}_t^{(\ell+k)} = \text{softmax}\left(\mathbf{H}_t^{(\ell+k)}\right), \quad \widetilde{\mathbf{Q}}_t^{(\ell+k)} = \text{softmax}\left(\widetilde{\mathbf{H}}_t^{(\ell+k)}\right) \tag{19}$$

where $\mathbf{P}_t^{(\ell+k)}$ and $\widetilde{\mathbf{Q}}_t^{(\ell+k)}$ are the original and perturbed token distributions at the target layer.

The KL divergence for a single token is (Hershey & Olsen, 2007):

$$D_{\text{KL}}\left(\mathbf{P}_t^{(\ell+k)} \| \widetilde{\mathbf{Q}}_t^{(\ell+k)}\right) = \sum_{j=1}^{D} \mathbf{P}_{t,j}^{(\ell+k)} \log \frac{\mathbf{P}_{t,j}^{(\ell+k)}}{\widetilde{\mathbf{Q}}_{t,j}^{(\ell+k)}} \tag{20}$$

Finally, we compute the mean KL divergence across all samples and tokens:

$$\bar{D}_{\text{KL}}^{(\ell,\ell+k)} = \frac{1}{NT} \sum_{i=1}^{N} \sum_{t=1}^{T} D_{\text{KL}}\left(\mathbf{P}_{i,t}^{(\ell+k)} \| \widetilde{\mathbf{Q}}_{i,t}^{(\ell+k)}\right) \tag{21}$$

where $N$ is the number of sample inputs.

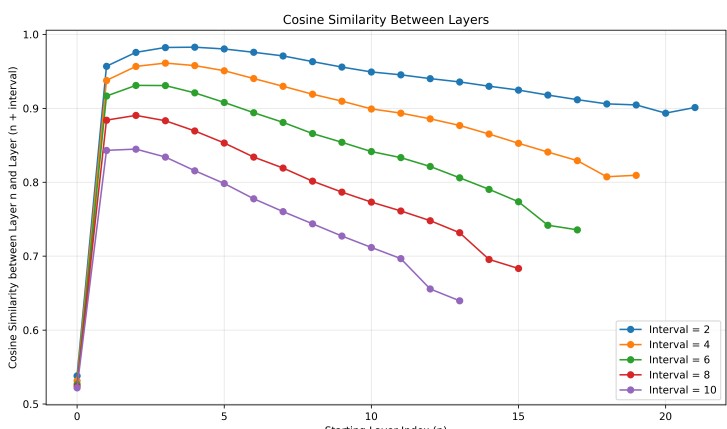

Figure 3: Cosine Similarity Distribution for OPT1.3B Model

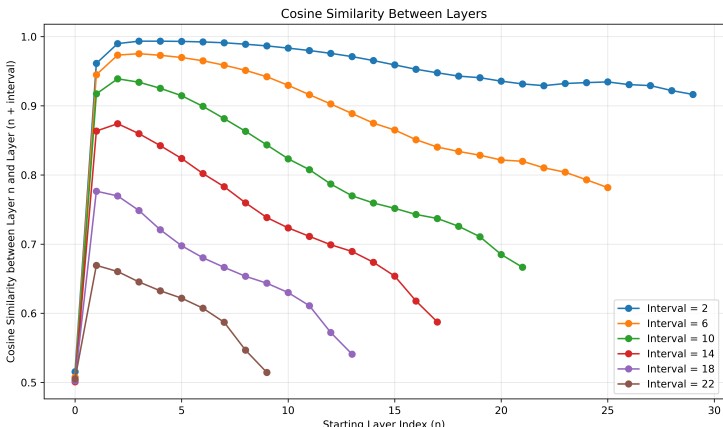

Figure 4: Cosine Similarity Distribution for OPT2.7B Model

A lower $\bar{D}_{\mathrm{KL}}^{(\ell,\ell+k)}$ indicates greater invariance of the downstream layer $\ell + k$ to perturbations introduced at layer $\ell$, suggesting potential redundancy or robustness. This insight guides our pruning strategy by identifying stable, potentially compressible layers without degrading the model's representational capacity.

### A.5 COSINE SIMILARITY, KL DIVERGENCE AND LLMGD VALUES DISTRIBUTION

#### A.5.1 COSINE SIMILARITY

We analyze the cosine similarity distributions between interval-pruned layers in the OPT-1.3B and OPT-2.7B models to better understand internal redundancy and structural diversity under varying pruning depths. Each curve in Figure 3 and Figure 4 represents the pairwise similarity among layers selected via regular-interval pruning at a given pruning depth.

**OPT-1.3B** For OPT-1.3B, cosine similarity curves generally follow a rising-then-falling pattern. With shallow pruning (2 or 4 layers), similarity remains high (0.9–0.98), indicating strong representational redundancy between pruned layers. As pruning depth increases to 6, 8, and 10 layers, similarity steadily declines, suggesting growing representational differentiation. This trend reflects a relatively uniform semantic transition across layers, implying that OPT-1.3B layers are structurally consistent and exhibit limited hierarchical specialization.

**OPT-2.7B** OPT-2.7B displays a more dramatic decay. While shallow pruning (2 and 6 layers) shows similarly high similarity to the results of OPT-1.3B, deeper pruning (10 layers and beyond)

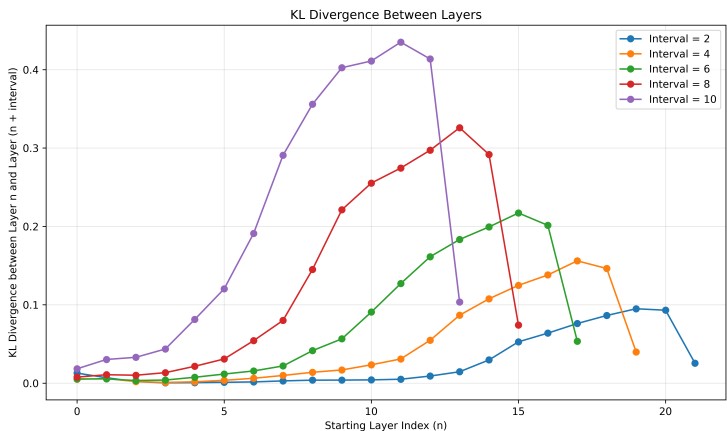

Figure 5: KL Divergence Distribution for OPT1.3B Model

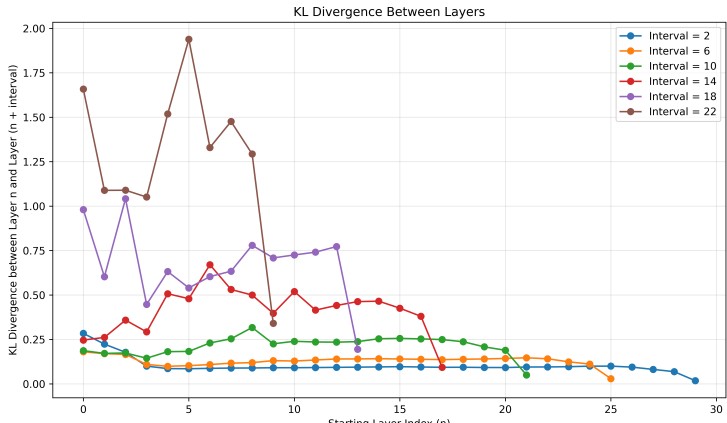

Figure 6: KL Divergence Distribution for OPT2.7B Model

yields significantly lower similarities; dropping to the range of 0.5–0.6 at 18 or 22 layers. This rapid decline suggests that deeper layers in OPT-2.7B are structurally and semantically more specialized. The pattern suggests the presence of internal modularity and a richer hierarchy of representations.

**Summary** The initial rise in similarity followed by decay reflects internal alignment among early layers and divergence as pruning deepens. OPT-1.3B's gradual decay contrasts with the sharper divergence in OPT-2.7B, highlighting architectural scaling effects. These observations offer practical implications for pruning strategy, indicating that interval-based layer selection is effective in the shallow regime, while deeper pruning may require recovery modules to maintain performance.

### A.5.2 KL DIVERGENCE

We further analyze the KL divergence between the hidden state distributions of interval-pruned layers under a specific perturbation to assess the functional dissimilarity across layer positions in OPT-1.3B and OPT-2.7B. Results are shown in Figure 5 and Figure 6, respectively.

**OPT-1.3B** In OPT-1.3B, KL divergence increases gradually with pruning depth. For shallow pruning (2 or 4 layers), divergence values remain below 0.1, indicating high semantic alignment and redundancy among pruned layers. As pruning deepens (6–10 layers), the divergence curves rise smoothly, without abrupt changes, suggesting progressive yet controlled representational diversification. This regularity implies that the internal layers of OPT-1.3B are relatively homogeneous and can tolerate perturbation to some extent, making it structurally suitable for uniform interval-based pruning.

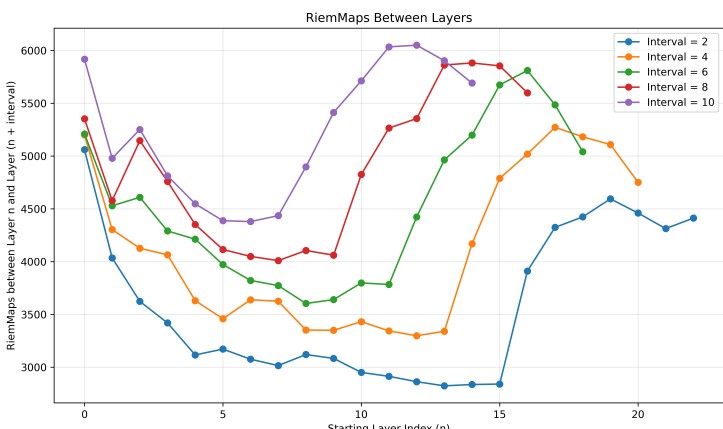

Figure 7: LLMGD Values Distribution for OPT1.3 Model

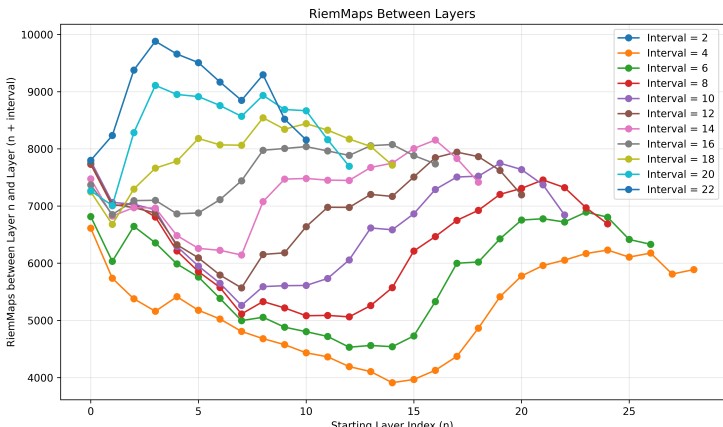

Figure 8: LLMGD Values Distribution for OPT2.7B Model

**OPT-2.7B**   By contrast, OPT-2.7B exhibits much higher and less stable divergence behavior. While shallow pruning (2 or 6 layers) produces relatively low KL values, deeper pruning (14–22 layers) introduces sharp peaks, with divergence values exceeding 1.5 in some cases. These abrupt changes indicate that deeper layers in OPT-2.7B are more functionally specialized and encode semantically distinct transformations. The irregularity of the curves further implies the presence of modular substructures and non-uniform transitions across the model depth.

**Summary**   KL divergence curves highlight the architectural contrast between models. OPT-1.3B shows smooth, interpretable divergence progression, while OPT-2.7B exhibits spike-prone, high-variance behavior under deep pruning.

### A.5.3   LLMGD Values

To further assess the geometric characteristics of hidden representations, we computed LLMGD values between pruned layer pairs in both OPT-1.3B and OPT-2.7B. These values quantify the intrinsic distance between layer representations in a nonlinear manifold space, and provide insights into how representational complexity evolves with depth. The results are shown in Figure 7 and Figure 8, respectively.

**OPT-1.3B**   LLMGD values in OPT-1.3B exhibit a U-shaped structure across nearly all pruning depths. Values first decrease, reaching a minimum around middle layers, then increase again as the pruned positions move deeper. This symmetric pattern implies that the model transitions from complex low-level processing (early layers), through a compressed semantic bottleneck (middle

layers), and back to complex task-specific representations (late layers). As pruning depth increases (e.g., 6 to 10 layers), the LLMGD curves skew upward, suggesting greater semantic diversity among selected layers.

**OPT-2.7B** In contrast, the LLMGD profiles of OPT-2.7B reveal more irregular and hierarchical structure. With deeper pruning (14–22 layers), the curves no longer follow a U-shape, but instead contain multiple local peaks. This pattern implies the presence of functional modules and discontinuities in the model's representational space. Notably, high LLMGD values in certain regions indicate that some layers are geometrically distant from others—highlighting their role as semantically distinct processing blocks.

**Summary** Compared to the smoother trajectories in OPT-1.3B, the irregular and high-curvature structures in OPT-2.7B reflect stronger specialization and internal modularity. These observations support the notion that large models develop deep hierarchical features and that pruning strategies should adapt to such representational stratification.

