# OpenReview forum: "LLMGD: Compressing LLMs with Layerwise Geodesic Distances"
_ICLR.cc/2026/Conference — ICLR 2026 Conference Withdrawn Submission_

### Official Review · Reviewer_Ucuh · 2025-10-28

**Soundness:** 2
**Presentation:** 3
**Contribution:** 2
**Rating:** 2
**Confidence:** 4

**Summary:**

This paper aims to address the compression problem of large language models (LLMs), specifically how to reduce computational costs through layer pruning. The authors point out that many existing pruning methods lack a solid theoretical foundation. To this end, the paper proposes a new, theoretically grounded pruning metric called LLMGD (Large Language Model Geodesic Distances).

The core idea of LLMGD is to quantify the representational distortion between adjacent layers in the model. If the LLMGD value (i.e., the geodesic distance) between two layers is small, it indicates low representational distortion, and the corresponding layer block can be regarded as structurally redundant and pruned. Furthermore, the paper theoretically establishes the relationship between LLMGD and the bi-Lipschitz constant, proving that LLMGD serves as an upper bound for representational distortion.

In the experimental section, the authors conduct pruning experiments on OPT-1.3B and OPT-2.7B models, replacing redundant layers with lightweight modules, and compare the results with baseline methods such as cosine similarity and KL divergence.

**Strengths:**

The paper introduces a new layer-wise metric, LLMGD, which estimates the layer-wise SPD precision matrix using a probabilistic graphical model and computes the geodesic distance on a Riemannian manifold to guide LLM pruning.

**Weaknesses:**

The weaknesses mainly focus on experimental validation.
1. Table 1 (OPT-1.3B) and Table 2 (OPT-2.7B) use different evaluation datasets, making direct comparison unreliable.
2. For multiple-choice tasks such as MMLU, model accuracy hovers around 25%, which is essentially random guessing; thus, these results fail to demonstrate the effectiveness of different pruning strategies.
3. The experiments only involve OPT-1.3B and OPT-2.7B. Although the paper claims future extension to larger models, this is unconvincing since newer models such as Qwen3-0.6B and Qwen3-1.7B are smaller yet significantly stronger; these should have been included for comparison.
4. While cosine similarity and KL divergence are considered, other key pruning baselines in recent literature are missing.
5. The study does not include experiments on generative tasks, which limits the generalizability of the proposed approach. [1][2]

[1] BlockPruner: Fine-grained Pruning for Large Language Models

[2] Shortened LLaMA: Depth Pruning for Large Language Models with Comparison of Retraining Methods

**Questions:**

See weakness

---

### Official Review · Reviewer_rQmr · 2025-10-28

**Soundness:** 2
**Presentation:** 2
**Contribution:** 2
**Rating:** 2
**Confidence:** 4

**Summary:**

This paper proposes LLMGD, a novel compression method for large language models (LLMs) based on Layerwise Geodesic Distances. The approach quantifies inter-layer representational geometric distortion by computing the geodesic distance between corresponding SPD matrices of adjacent layers, thereby identifying and pruning structurally redundant layers.

However, the theoretical interpretation of LLMGD remains insufficiently developed. Although the paper establishes a theoretical connection between the value of LLMGD (i.e., the inter-layer geodesic distance) and the bi-Lipschitz constant, it lacks a deeper explanation of how LLMGD inherently captures structural redundancy and how well this mechanism generalizes across different models and tasks.

Moreover, the current experiments only evaluate the method on classification benchmarks using OPT-1.3B and OPT-2.7B models. The approach still lacks validation across broader model families, larger scales, and generation tasks such as reasoning, code generation, and long-text understanding.

**Strengths:**

- The paper introduces the geodesic distance as a metric to quantify geometric changes between layer representations, which allows it to capture more complex nonlinear geometric distortions compared to methods based on cosine similarity or KL divergence.
- The proposed LLMGD method leverages the Affine-Invariant Riemannian Metric (AIRM) on Symmetric Positive Definite (SPD) matrices and establishes a theoretical connection with the bi-Lipschitz constant, providing a solid mathematical foundation for assessing the geometric stability of layer transformations.
- On classification tasks using OPT-1.3B and OPT-2.7B models, LLMGD outperforms pruning schemes based on Cosine Similarity and KL Divergence across multiple datasets.

**Weaknesses:**

1. Although the paper theoretically connects the LLMGD value (inter-layer geodesic distance) to the bi-Lipschitz constant, showing that LLMGD can measure the geometric stability of layer transformations, it only establishes a correlation between geometric properties (low distortion) and pruning decisions (removability) without probing the causal mechanism behind this link.
2. There is a lack of key hyperparameter specification and analysis: the paper neither reports the exact values used nor studies their impact. For example:
  - The k value in the k-NN graph
  - The β regularization parameter in Graphical Lasso
  - The architecture and training settings of the lightweight replacement layer
3. Experiments are conducted only on OPT models. Different architectures can have very different internal geometries and information flows. Is the geometric distortion measured by LLMGD still a reliable indicator of redundancy across other architectures and model scales?
4. The method is evaluated only on classification benchmarks, with no assessment of generative tasks. Classification is often less sensitive to fine-grained representational nuances than generation.
5. The method’s MMLU scores are below random guess. The baselines (OPT-1.3B/2.7B) exhibit little knowledge/reasoning ability on MMLU (a 4-choice task where random is ~25%). If the original models lack this ability, the pruning study cannot meaningfully assess whether LLMGD better preserves it than competing methods.
6. Missing ablations:
  - Component contributions in LLMGD: The pipeline comprises several steps (e.g., PGM/precision-matrix estimation, geodesic computation). There is no ablation quantifying each component’s contribution (e.g., what if a different precision-matrix estimator is used?).
  - Effect of the replacement layer: The paper uses a lightweight replacement module to substitute pruned layers. Please compare against pruning without a replacement to isolate the module’s benefit.
7. Incomplete baselines vs. recent pruning advances, such as:
  - ShortGPT: Layers in large language models are more redundant than you expect (Men et al., 2024)
  - SLEB: Streamlining LLMs through redundancy verification and elimination of transformer blocks (Song et al., 2024)
  - LLM-Pruner: On the structural pruning of large language models (Ma et al., 2024)
  - SliceGPT: Compress large language models by deleting rows and columns (Ashkboos et al., 2024)
  - LaCo: Large language model pruning via layer collapse (Yang et al., 2024)

**Questions:**

See weakness

---

### Official Review · Reviewer_qhwy · 2025-10-29

**Soundness:** 1
**Presentation:** 2
**Contribution:** 2
**Rating:** 2
**Confidence:** 3

**Summary:**

The paper “LLMGD: Compressing LLMs with Layerwise Geodesic Distances” introduces a geometry-based approach to analyze and compress large language models (LLMs). The authors propose a pruning framework that quantifies representational change across layers by computing geodesic distances between precision matrices estimated from hidden states. Using probabilistic graphical models and the affine-invariant Riemannian metric on symmetric positive definite (SPD) matrices, LLMGD identifies layers that cause minimal geometric distortion—interpreted as redundant layers suitable for removal or lightweight replacement. The method is theoretically supported through a bi-Lipschitz upper bound interpretation and empirically shown to outperform existing pruning metrics on models such as OPT-1.3B and OPT-2.7B, achieving improved compression–accuracy trade-offs.

**Strengths:**

- Originality: This paper introduces a new method to evaluate how information shifts from layer to layer which has never been introduced before, based on computing geodesic distances between precision matrices estimated from internal reps of LLMs. In particular, the use of probabilistic graphical models on internal reps of LLMs is novel.
- Quality: The techniques explained draw from mathematically grounded concepts. Computational expense is kept feasible despite several heavy operations are involved.
- Clarity: The idea is explained clearly and the strategy is illustrated in depth throughout the paper.
- Significance: the effort of reducing LLM is certainly timely and important.

**Weaknesses:**

I have a few objections about this work:
- Broad comment: the authors repeat frequently that in the business of understanding how LLMs layers process and distort information, previous work is mostly empirical and not theoretically grounded. While this might be true (with exceptions), it is hard for me to justify how this work takes a theoretically grounded approach to this problem. It seems to me that, instead, the authors take a mathematically grounded technique to evaluate information processing through layers, using these evaluations to prune layers, which is not the same as addressing the theoretical problem of how information flows through layers. I might have misunderstood, more the reason for the authors to clarify this point as a way of improving their manuscript.
- The set of experiments is limited to small-medium models (and only one model really, OPT), while existing literature (cited in the work) has results on larger models. I believe the authors should compare more closely to existing literature (e.g. Men et al, Gromov et al.) with similar models and setups, to prove their method is better performing.
- Mean pooling over the token dimensions seems not the standard choice for causal LLMs, given the last token has most of the information. The authors should consider changing this choice, or motivating why they use mean pooling.
- In experiments, the authors quote results for MMLU benchmark, for which the models they consider already performs basically random choice before pruning. This is testing the method in an unstable limit where results do not make sense (after pruning, the model performs worst than random choice?)
- Relevant work section seems to be limited to probabilistic graphical models, while literature on pruning LLMs is mentioned a bit sparsely (and with some omissions, e.g. Zhang et al. 2024, Kim et al. 2024, Gardinazzi et al. 2025 just to name a few).

**Questions:**

I will formulate more precise questions along the lines of what written above:
- Can the authors run a set of experiments to directly compare to previous literature (e.g. Gromov et al, Men et al)?
- Can the authors run a quick experiment to compare the current pipeline to choosing last token representation over mean pooling?

---

### Official Review · Reviewer_qPJo · 2025-11-01

**Soundness:** 2
**Presentation:** 3
**Contribution:** 2
**Rating:** 4
**Confidence:** 4

**Summary:**

The paper proposes LLMGD, a geometry based method for pruning layers in LLMs. The main idea is to look at each layer through the covariance structure of its hidden representations and to measure the change between consecutive layers by using the affine-invariant Riemannian metric (AIRM). Layers that induce only a small geometric change are considered redundant and are removed. To mitigate significant accuracy loss, a lightweight replacement layer is trained on input/output pairs generated from the original model. On OPT-1.3B/2.7B, across several benchmarks (BoolQ, PIQA, COPA, ARC-Easy) LLMGD usually outperforms cosine and KL based pruning at similar pruning ratios. The method is data-driven, but some of its drawbacks are the computational cost of estimating precision matrices/eigendecompositions and that the experiments are limited to mid-sized, mainly classification style evaluations.

**Strengths:**

- Experiments on multiple tasks and two OPT model sizes show gains over cosine/KL.
- The pipeline includes a recovery step where a lightweight layer is trained to reduce the performance loss.
- The theoretical framing with a bi-Lipschitz argument gives the approach a clear justification.
- The approach targets structured (layer-level) compression, which is better for deployment than unstructured sparsity.
- The method is data driven, deriving pruning decisions from actual layer activations rather than simple static weight statistics.

**Weaknesses:**

- The related work section is quite short and does not clearly position the method among structured/unstructured pruning, depth-adaptive models, or recent data based criteria.
- The method does not always outperform baselines on every task and pruning level.
- The approach depends on being able to estimate precision matrices and being able to do eigen decompositions, which becomes significantly expensive as the sample size grows (Table 3) thus limiting the practicality for large-scale use-cases.
- The evaluation is restricted to mid-sized OPT models and mainly classification based benchmarks, so it is not clear how well the method would transfer to larger, instruction-tuned, or generative settings.
- The pruning signal is explored together with a retraining/replacement step, so it is hard to understand how strong the geometric criterion is by itself.

**Questions:**

- Please expand the related work to make sure it covers structured vs unstructured pruning as well as depth-adaptive methods, and recent data-driven pruning criteria while clearly stating where LLMGD fits.
- Report aggregate metrics (average accuracy) for Tables 1-2 so the overall effect is clear despite per-task variance.
- Clarify the computational setup for Table 3 (hardware, per-layer cost, number of pairs) and compare it to simpler signals like cosine/KL.
- Add generation based evaluations (e.g. perplexity) to show the method transfers beyond classification on OPT and also expand to a different family of models.
- Separate the effect of the geometric signal from the retraining step by reporting accuracy right after pruning as well, before the replacement.

I'd be happy to increase my score if these points are resolved!

---

### Note · Authors · 2025-11-19

I have read and agree with the venue's withdrawal policy on behalf of myself and my co-authors.